# Production of Recombinant EAV with Tagged Structural Protein Gp3 to Study Artervirus Minor Protein Localization in Infected Cells

**DOI:** 10.3390/v11080735

**Published:** 2019-08-09

**Authors:** Anna Karolina Matczuk, Grzegorz Chodaczek, Maciej Ugorski

**Affiliations:** 1Department of Pathology, Division of Microbiology, Faculty of Veterinary Medicine, Wrocław University of Environmental and Life Sciences, Wrocław 50-375, Poland; 2Department of Biochemistry and Molecular Biology, Faculty of Veterinary Medicine, Wrocław University of Environmental and Life Sciences, Wrocław 50-375, Poland; 3Confocal Microscopy Laboratory, PORT Polish Center for Technology Development, Wrocław 54-066, Poland

**Keywords:** EAV, arterivirus, Gp3, tagged recombinant virus, co-localization, trafficking

## Abstract

Equine arteritis virus (EAV) is a prototype member of the Arterivirus family, comprising important pathogens of domestic animals. Minor glycoproteins of Arteriviruses are responsible for virus entry and cellular tropism. The experimental methods for studying minor Arterivirus proteins are limited because of the lack of antibodies and nested open reading frames (ORFs). In this study, we generated recombinant EAV with separated ORFs 3 and 4, and Gp3 carrying HA-tag (Gp3-HA). The recombinant viruses were stable on passaging and replicated in titers similar to the wild-type EAV. Gp3-HA was incorporated into the virion particles as monomers and as a Gp2/Gp3-HA/Gp4 trimer. Gp3-HA localized in ER and, to a lesser extent, in the Golgi, it also co-localized with the E protein but not with the N protein. The co-localization of Gp3-HA and the E protein with ERGIC was reduced. Moreover, EAV with Gp3-HA could become a valuable research tool for identifying host cell factors during infection and the role of Gp3 in virus attachment and entry.

## 1. Introduction

Equine arteritis virus (EAV) is a prototype member of *Arteriviridae*, a family of enveloped positive-stranded RNA viruses comprising porcine reproductive and respiratory syndrome virus (PRRSV), a major pathogen in the swine industry [1]. EAV infects horses and donkeys and leads to abortions in pregnant mares and respiratory illness with flu-like symptoms, which can even lead to death in young animals. The virus is transmitted via the respiratory route and via the contaminated semen of previously infected stallions. Despite available vaccines, EAV remains an important pathogen in the horse industry [2].

The infectious genomic RNA has a length of ~12 kbp, is 3′-polyadenylated, and presumably 5′-capped. The large replicase open reading frames (ORFs) 1a and 1b occupy most of the EAV genome and are directly translated from the genomic RNA [1,3]. However, the genes for the structural virion proteins, which are located at the 3’-end of the arterivirus genome, overlap with each other and are expressed from the 3’ co-terminal nested set of six leader-containing subgenomic RNAs [4].

The structural proteins of arteriviruses include the nucleocapsid N and several membrane proteins, such as the Gp5/M dimer, the Gp2/3/4 complex, the small and hydrophobic E protein, and the ORF5a protein [5]. All structural proteins, in addition to ORF5a, are essential for EAV infectivity; however, only N along with viral RNA and Gp5/M dimer is required for budding. Note that Gp2, Gp3, Gp4, and E are minor viral components, and knocking them out individually from EAV does not prevent the budding of non-infectious virus-like particles (VLPs) [6]. If one of the genes encoding Gp2, Gp3, or Gp4 is deleted, the produced VLPs do not contain any Gp2/3/4 trimer component and have reduced the amount of the E protein. If the E protein is knocked out the Gp2/3/4 trimer, it is not incorporated into the virion particles, suggesting the involvement of E in the assembly of infectious particles containing the trimer [6]. The trimer itself serves as a major tropism determinant for arteriviruses. The chimeric PRRSV arterivirus, containing E and Gp2/3/4 exchanged to homolog proteins of EAV, exhibits broadened cellular tropism of the EAV [7]. In PRRSV, the Gp2/3/4 was shown to attach to the CD163 cellular receptor on the porcine alveolar macrophages [8]. Recently, genome-edited pigs, lacking part of the CD163 receptor, were shown to be resistant to the PRRSV challenge [9]. E is a small, hydrophobic protein suspected to serve as a viroporin [10]. The Gp2 and Gp4 are type I integral membrane proteins [5]. The Gp3 is a peculiar protein, which was shown to form a disulfide-linked trimer with only Gp2/4 in the virus particles, and the trimer cannot be detected in the infected cells [11].

Intracellular localization of the minor proteins of EAV and other arteriviruses has not been studied in detail because of the problems associated with production of antibodies against these highly glycosylated proteins. In infected cells the E protein localizes mostly in the ER, and some fraction of it is presumably located in the Golgi, as it was shown to co-localize with Gp5 [12]. Both Gp3 and Gp4, which were individually expressed (transfection with vaccinia system), were shown to co-localize with concanavalin A, an ER marker [13]. It is not known wheather the Gp3 localizes beyond the ER and what is its localization in infected cells.

The location for arterivirus budding is not known. Previous studies have shown that arterivirus budding involves the wrapping of a preformed nucleocapsid by membranes in the ER or Golgi complex [14,15]. Moreover, infection with arteriviruses resulted in the re-arrangements of the internal membranes and formation of large numbers of double membrane vesicles (DMVs). For the formation of DMVs, non-structural proteins nsp2 and nsp3 are sufficient [16].

In this study, we created two recombinant EAV viruses: one with separated ORF3 and ORF4, and one with Gp3 and C-terminus HA-tag. The mutant viruses exhibited similar growth proprieties. Gp3-HA was incorporated into the virions and formed multimeric complexes, corresponding to the Gp2/3/4 trimer size. The EAVGp3-HA mutant enabled us to localize Gp3-HA using secretory pathway markers and other EAV proteins. We have shown that Gp3-HA have similar distribution as the E protein, i.e., it mainly localize in the ER, while some of it is moved to cis-Golgi. Finally, we demonstrated that the arterivirus genome tolerates the separation of the ORFs in virion structural genes and that the small tag can be introduced into Gp3 without loss in infectivity of the EAV virus. These results can be implemented for the development of new arteriviral vaccines.

## 2. Materials and Methods

### 2.1. Cells and Media

The cell line BHK-21 (baby hamster kidney cells; ATCC C13) was maintained as an adherent culture in DMEM mixed in a 1:1 ratio with the Leibovitz L-15 medium (Cytogen, Lodz, Poland), which was supplemented with 5% fetal calf serum (FCS) (Biological Industries, Cromwell, IA, USA), 100 U of penicillin per mL, and 100 mg of streptomycin and l-glutamine per mL (Biowest, Lodz, Poland). The cells were maintained at 37 °C in an atmosphere of air with 5% CO_2_ and 95% humidity.

### 2.2. Generation of Mutant EAV Genomes

Recombinant DNA techniques were performed according to standard protocols. Newly constructed plasmids were propagated in competent *Escherichia coli* strain DH5alpha (Thermo Scientific, Warszawa, Poland). Cloning vector pEAV211, which is a derivate of the pEAV030 (Genbank Y07862.2) and described in reference [17], was used to generate mutant EAVs. We generated specific DNA fragments by overlap extension PCR and purified them from the gel with the aid of the Gel-out kit (A&A Biotechnology, Gdansk, Poland). RGEAVBamHI and RGEAVEcoRI were used as an external primers. First, the translation initiation codon of the EAV ORF4 was mutated (ATG > ACG) with primers Gp4KOFor and Gp4KORev, and the 2.345 bp fragment cloned back to pEAV211 with the restriction enzymes *BamHI* and *EcoRI* (Thermo Scientific, Poland) to generate pEAV211Gp4KO. In the second step, the *AscI* site was added after the stop codon of the ORF3 with primers EAV211AscIFor and EAV211AscIRev to generate the plasmid pEAV211Gp4KOAscI. The separation of the ORF3 and ORF4 was achieved by the overlap extension PCR with primers ReconAscGp4For and RGEAVEcoRI on the pEAV211 template and cloned to pEAV2114KOAscI with the *AscI* and *EcoRI* restriction enzymes to generate pEAV211s3/4. As the last step, the HA tag was added directly to ORF3’s 3’-end with the use of primers RGEAVBamHIIFor and EAVGp3HARev and pEAV211s3/4 as a DNA template. The ligation after *BamHI* and *AscI* restriction enzyme digestion of the 1657 bp product produced the pEAV211Gp3-HA vector. All of the generated plasmids were sequenced with the RGEAVEcoRIRev primer (Genomed, Warszawa, Poland). Table 1 shows the list of used primers, while Figure 1 shows the cloning schematics. All the genes that were subjected to mutations in plasmids were sequenced before use in experiments (Genomed, Warsaw, Poland).

### 2.3. RNA Transcription and Generation of Mutant EAVs

Full length clones pEAV211, pEAV211s3/4, and pEAV211Gp3-HA were linearized using *XhoI* and in vitro-transcribed using AmpliCap-Max T7 High Yield Message Maker Kit (Cellscript, Madison, WI, USA), and 6 µg RNA was then introduced into the BHK-21 cells suspended in PBS using the Gene Pulser Xcell electroporation apparatus and electroporation cuvettes with a 4-mm electrode gap (Bio-Rad, Warszawa, Poland). The cells were pulsed twice at 850 V, 25 F; resuspended in DMEM/L-15 5% FCS; and seeded into two wells of the 6-well plate. The cells were then maintained at 37 °C until the CPE was observed. The cells that adhered were detached using a plastic cell scraper and collected together in the supernatants. The cells were then centrifuged at a low speed. While half of the cells were subjected to RT-PCR and sequencing, the second half were subjected to western blotting with anti-N and anti-HA antibodies. The remaining supernatants were collected, aliquoted, and stored in −80 °C as a P0 stock.

### 2.4. In Vitro Growth Characteristics of Generated EAV Mutants

The monolayers of BHK-21 cells grown in 6-well plates were inoculated with each of the wild-type EAV-WT (derived from pEAV211), EAVs3/4, and EAVGp3-HA viruses at a multiplicity of infection (MOI) of 0.1 and incubated at 37 °C for 2 h. The cells were then washed two times with PBS, with calcium and magnesium, and then overlaid with 2 mL of DMEM/L-15 1% FCS and 1% l-glutamine culture medium. At 6, 12, 24, 48, and 72 h post-infection, the supernatants were harvested and virus titers were determined on the BHK-21 cells using the plaque assay. Virus aliquots were stored at −80 °C. This experiment was carried out in triplicate.

### 2.5. Plaque Assay

Plaque assay was performed on the BHK-21 cells grown on 6-well plates with GMEM supplemented with 1% FCS, 1% L-glutamine, and 0.75% carboxymethyl cellulose (CMC, Sigma-Aldrich, Poznań, Poland). The overlays were fixed with 10% formaldehyde and stained with crystal violet for three days p.i.

### 2.6. Stability of the HA-Tag

To determine the stability of the HA-tag fluorescence, the recombinant EAVGp3-HA virus was subjected to 19 sequential serial passages at an MOI of 1 in the BHK-21 cells. After the appearance of CPE, the supernatants were collected, centrifuged at a low speed, and stored in −80 °C. The remaining cells were washed with PBS, centrifuged, and stored in −80 °C for further RT-PCR and sequencing. To verify the stability of the HA-tag, the BHK-21 cells were grown on glass coverslips that were placed on a 24-well plate and infected with different passage recombinant viruses at an MOI of 1. 18h p.i. the cells were subjected to immunofluorescence with anti-HA tag antibodies (1:500, ab9110, Abcam, UK) and anti-N antibody (1:250, VMRD, USA), as has been described later in the Materials and Methods section. Furthermore, the infected cells from passages P7, P10, P15, and P19 were subjected for RT-PCR and sequencing.

### 2.7. RT-PCR and Sequencing

The total cellular RNA was extracted from transfected or infected cells (from 6-well plates each) with RNeasy Mini kit (Qiagen, Wrocław, Poland) according to manufacturer’s instructions. The cDNA was generated with Maxima H minus First Strand cDNA synthesis kit (Thermo, Poland) according to the manufacturer’s instructions with 2 µg of RNA and oligoT primer. The obtained cDNA was subjected to PCR reaction with EAVFor and EAVRev primers (each 10 mM) listed in Table 1 and one-fusion high-speed-fidelity polymerase (GeneON, Abo, Poland). The thermal profile was as follows: initial denaturation at 98 °C for 5 min, followed by 35 cycles of denaturation at 98 °C for 10 s, annealing at 46 °C for 20 s and extension at 72 °C for 20 s, and a final extension at 72 °C for 3 min. The RT-PCR products were gel-purified using a QIAquick Gel Extraction Kit (QIAGEN Inc., Valencia, CA, USA), and the sense and antisense strands were sequenced (Eurofins MWG Operon, Huntsville, AL). The sequence data were analyzed using the software FinchTV 1.5.0. PCR products were subjected to agarose gel electrophoresis, purified (Gel-Out, A&A Biotechnology, Gdynia, Poland), and the sense and antisense strands were sequenced (Genomed, Warsaw, Poland).

### 2.8. Analysis of the Gp3-HA Expression from Infected Cells

The BHK-21 cells seeded on 6-well plates were infected with P1 stock at an MOI of 1 or left mock infected. 18 h p.i. cells were detached using a plastic cell scraper, pelleted, and lysed either directly in the SDS sample buffer without DTT or resuspended in 80 µL 1× glycoprotein denaturing buffer and boiled for 10 min at 100 °C. To analyze glycosylation, the samples were digested with peptide-N-glycosidase (PNGase F; 5 U/L, 2 h at 37 °C) or endo-beta-*N*-acetyl-glucosaminidase (Endo H; 5 U/L, 1 h at 37 °C) according to the manufacturer’s instructions (New England BioLabs, Hitchin, UK). After the deglycosylation reaction, the samples were supplemented with reducing SDS-PAGE buffer and subjected to SDS-PAGE and western blotting.

### 2.9. Immunofluorescence Assay

The BHK-21 cells were seeded in the complete medium onto glass coverslips in 24-well plates. After 24 h, the cell culture medium was replaced with DMEM/L-15 medium with the cells infected with EAV wt, EAV s3/4, EAVGp3-HA, or left uninfected. After 2 h, the cells were washed two times with PBS and the cell culture medium was replaced with DMEM/L-15 containing 1% FCS, 1% penicillin/streptomycin and, 1% l-glutamine. The cells were then fixed 18 h p.i. with paraformaldehyde (4% in PBS) for 15 min at room temperature (RT), washed twice with PBS, permeabilized with 0.5% Triton in PBS for 3 min at room temperature, and washed again twice with PBS. After blocking (blocking solution contained 3% bovine serum albumin in PBST) for 1 h at room temperature, the cells were incubated with a rabbit polyclonal anti-HA tag antibodies (1:500, ab9110, Abcam, Cambridge, UK) and mouse monoclonal anti-N (1:250, VMRD, Pullman, WA, USA) antibody diluted in a blocking solution at room temperature for 1 h. These cells were then washed three times with PBS and incubated with secondary antibodies (1:800, goat anti-mouse IgG H&L Alexa Fluor 568 and 1:800 goat anti-rabbit IgG H&L Alexa Fluor 488, Abcam, UK). After immunostaining in all the cases, the cells were washed three times with PBS. Finally, the stained cultures were mounted on glass slides in a fluoroshield mounting medium with DAPI (Abcam, Cambridge, UK) and stored at 4 °C.

The images were recorded using a Zeiss Cell Observer SD confocal microscope (Zeiss, Oberkochen, Germany) equipped with an EMCCD QImaging Rolera EM-C^2^ camera and 40–63× oil objectives (0.167 μm and 0.106 μm per pixel, respectively). The imaging was performed sequentially using 405 nm, 488 nm, and 561 nm laser lines with a quadruple dichroic mirror 405 + 488 + 561 + 640 and 450/50, 520/35, and 600/52 emission filters. Subsequently, the images were deconvoluted with Huygens software (SVI, Hilversum, The Netherlands).

### 2.10. Co-Localization Assay

For localization with cellular markers, the BHK-21 cells were infected with EAVGp3-HA at an MOI of 1 or left uninfected. The cells at 18 h p.i were subjected to immunofluorescence as described above with the following antibodies: rabbit polyclonal anti-HA tag antibodies (1:500, ab9110, Abcam, UK), mouse monoclonal anti-HA tag antibody (1:200, 16B12 Enzo Life Sciences, Farmingdale, NY, USA), mouse monoclonal anti-N antibody (1:250, VMRD, USA), rabbit anti-E antibodies, mouse monoclonal anti-PDI (1:250, 1D3, Enzo Life Sciences, USA), mouse monoclonal anti-membrin (1:100, 4HAD6, Enzo Life Sciences, USA), and mouse monoclonal anti-ERGIC (1:150, OTI1A8, Enzo Life Sciences, USA). Each co-localization experiment was conducted at least 3 times, and at least 10 cells were taken to quantify the co-localization in the JACoP plugin using the Fiji software.

### 2.11. Computer Analysis

In this study, all the graphs presented were created using GraphPad Prism 8 software. The significance was estimated using one-way ANOVA. The images were obtained from the confocal microscope were deconvoluted using Huygens Essential X11 (Scientific Volume Imaging, Hilversum, The Netherlands) and processed in ImageJ Fiji [18]. The co-localization analyses were performed in ImageJ using the JACoP plugin [19] in which Pearson’s and Mander’s coefficient were calculated. For each condition, at least 10 cells were taken for measurements of co-localization. Fluorescence intensities in the antibody accessibility experiment were measured as a mean gray value in ImageJ Fiji for 10 fields per condition.

### 2.12. SDS-PAGE and Western Blotting

The detached and low-speed centrifuged, transfected, or infected cells were solubilized in the RIPA buffer (Sigma-Aldrich, Poland) with the complete protease inhibitor tablet (Roth, Sigma-Aldrich, Poland/Merc, Poland). Samples in the SDS-PAGE loading buffer, with or without DTT, were subjected to SDS-PAGE using 15% or 12% polyacrylamide. Then, the gels were blotted onto polyvinylidene difluoride (PVDF) membrane (GE Healthcare, Warszawa, Poland). After blocking of the membranes (blocking solution; 5% skim milk powder in PBS with 0.1% Tween 20 (PBST)) overnight at 4 °C, the antibodies in the blocking solution were incubated for 1.5 h at room temperature. Rabbit- anti-HA tag antibodies (1:6000); ab9110; Abcam, Cambridge, UK) were used to detect Gp3 with the HA tag, mouse monoclonal anti-N antibody (1:4000, VMRD, USA), and rabbit anti-E (1:1000, described in [12], a gift from Eric Snijder, University of Leiden, Belgium). After washing (3 times for 10 min each with PBST), suitable horseradish peroxidase-coupled secondary antibodies (1:4000; anti-rabbit or anti-mouse; Dako, Carpinteria, CA, USA) were applied for 1 h at room temperature. After washing with PBST, the signals were detected by chemiluminescence using the ECL plus reagent (Pierce/Thermo, Poland).

### 2.13. Immunoprecipitation

The BHK-21 cells seeded in T175 bottles were infected with EAVGp3-HA or mock infected at an MOI of 1. Two hours after infection cells were washed with PBS, and the cell culture medium was replaced with DMEM/L-15 containing 1% FCS, 1% penicillin/streptomycin and, 1% l-glutamine. 22 h p.i. supernatants were collected and low-speed centrifuged to remove the cells. Then, the supernatants were subjected to purification and concentration on Amicon Ultra-15, Ultracel-100K filters (Merck, Warszawa, Poland). Briefly, 20 mL of cell-free supernatants were transferred to filters and centrifuged at 3500× *g* for 40 min at room temperature, to obtain 100 uL of concentrated supernatant. The SDS sample buffer with or without DTT was added to 5% of the volume of the obtained concentrated virions, while the remaining 95% was lysed in the MNT buffer (20 mM MES, 30 mM Tris, 100 mM NaCl, 1% TX-100, pH 7.4) and subjected to IP with rabbit polyclonal anti-HA (ab9110; Abcam, Cambridge, UK) overnight at 4 °C. The antibody-protein complexes were pulled with A-Sepharose (Sigma-Aldrich, Poland), washed with MNT, boiled with reducing or non-reducing SDS buffer, and subjected to SDS-PAGE and western blotting as described above.

## 3. Results

### 3.1. Construction of the Recombinant EAV with Gp3 Carrying HA-Tag (Gp3-HA)

For recombinant EAV with a tagged Gp3, we selected the junction between ORF3 and ORF4 because, in the pEAV211 vector, the overlap between the ORF3 and ORF4 consists of 98 nt, while that between ORF2 and ORF3 is 202 nt. The ORF4 3’-end overlaps with two ORFs coding Gp5 and ORF5a proteins. Moreover, joining the HA-tag directly with the Gp3 did not abolish the expression of Gp3 [20], while the direct tagging of Gp2 with FLAG-tag made its expression impossible [21]. The cloning procedure is shown in Figure 1A. First, the start codon of Gp4 was mutated to produce the Gp4 knock-out virus, and then the restriction site *AscI* was generated to allow the separation of ORF3 and ORF4 by duplicating the missing ORF4 98 nt long sequence. Finally, the HA-tag sequence was directly inserted to the end of ORF3. All these mutations were generated using PCR methods. Subsequently, the in vitro-transcribed full-length viral RNAs from the *XhoI*-linearized plasmids pEAV211, pEAV211s3/4, and pEAV211Gp3-HA were introduced into the BHK-21 cells using electroporation. After 48 h, the cells were harvested and examined by western blotting analysis using antibodies that are specific to the N protein and HA-tag. The expression of Gp3-HA was only observed in cells transfected with in vitro-transcribed RNA produced from pEAV211Gp3-HA; however, it was not observed in pEAV211 (wt), pEAV211s3/4, or mock-transfected cells. Note that the size of the Gp3-HA was ~35 kDa (Figure 1B). As shown in Figure 1B, which shows the successful replication of the EAV, the expression of the N protein was observed in cells transfected with in vitro-transcribed RNA produced from pEAV211 (wt), pEAV211s3/4, and pEAV211Gp3-HA, but not in the mock-transfected cells.

When new cultures of the BHK-21 cells were infected using culture supernatants from BHK-21 cells previously transfected with in vitro-transcribed RNA from pEAV211 (wt), pEAV211Gp3-HA, or pEAV211s3/4, i.e., passage 0 (P0), Gp3-HA was detectable using only immunofluorescence (IF) for the EAVGp3-HA infected cells (24 h post-infection) (Figure 1C). However, the viral N protein was detected in cells infected with wild-type EAV (EAV-wt), EAVs3/4, and EAVGp3-HA, but not in mock-infected cells. To summarize, these data demonstrate that RNAs derived from pEAV211s3/4 and pEAV211Gp3-HA viruses are fully replication competent when introduced into the susceptible mammalian cells. In addition, the progeny viruses that have either separated ORF3 and ORF4 or express HA-tagged Gp3 are infectious.

### 3.2. In Vitro Growth Characteristics of EAVs3/4, EAVGp3-HA, and EAV-wt

The in vitro growth properties of EAVs3/4 and EAVGp3-HA viruses were compared to those of EAV-wt using one-step growth curve experiments for BHK-21 cells (Figure 2A). All tested viruses replicated to titers that exceeded 10^7^ PFU/mL of cell culture fluid, although the EAV-wt reached a slightly higher titer at 48 h post-infection. This suggests that the separation of the ORF3 and ORF4, as well as introduction of the HA-tag to the Gp3 protein did not have a significant effect on the EAV progeny’s production.

### 3.3. The Recombinant EAVGp3-HA Virus is Stable upon Replication

The stability of Gp3-HA expression was investigated by a serial passage of EAVGp3-HA virus in the BHK-21 cells. The expression of Gp3-HA was analyzed by IF using cells infected with P7, P10, P15, and P19 (Figure 2B). The levels of N and Gp3-HA protein expression was stable up to P15; however, in cells infected with P19, the number of cells expressing Gp3-HA and fluorescence intensity were reduced. To determine whether these results were caused by the emergence of viruses containing mutations, the intracellular RNA was isolated after infection with P7, P10, P15, and P19 EAVGp3-HA virus and subjected to RT-PCR, followed by sequence analysis (Figure 2C). Note that the consensus sequence was still present in P19; however, the chromatogram of the nt sequence of the HA-tag and the neighboring sequence showed the presence of mutated nucleotides, which may indicate the rise of the quasispecies viruses with accumulated mutations within the HA-tag sequence. Such quasispecies of viruses, which are present in the supernatant along with the original recombinant virus, can lead to diminished fluorescence intensity and loss of fluorescence from the HA-tag in certain infected cells. To summarize, this data indicate that the separation of ORF3 and ORF4 and introduction of Gp3-tag can be stably accommodated within the EAV genome and that the HA-tag added to C-terminus of Gp3 is expressed at least up to P15.

### 3.4. The Gp3-HA in Infected Cells is Heterologously N-Glycosylated and Does not Pass the Medial Golgi Compartment

To investigate the molecular weight of Gp3-HA, the BHK-21 cells were infected with the EAVGp3-HA virus. Cell lysates were subjected to both non-reducing and reducing SDS-PAGE, followed by western blotting with anti-HA antibodies. In the non-reducing conditions, most of the Gp3-HA protein was present at the band of ~35 kDa, and only a small fraction migrated to two faint bands of ~50 kDa (Figure 3A). This indicates that the Gp3-HA in infected cells is present mostly in the monomeric form; however, some of it forms a multimeric structure that might correspond to the Gp3-HA dimer or the Gp3-HA associated with other viral or cellular proteins. In reducing conditions, the Gp3-HA displays a characteristic double-band pattern, which is caused by the heterologous glycosylation of the overlapping NNTT sequon, indicating that certain Gp3-HA molecules have 5 and some have 6 *N*-glycans [20]. To confirm that Gp3-HA in infected cells is indeed N-glycosylated, the cell lysates were digested with peptide N-glycosidase F (PNGase F) (cleaves of all types of N-linked carbohydrates) or endo-β-*N*-acetyl-glucosaminidase (Endo H; cleaves only the high-mannose-type carbohydrates) prior to SDS-PAGE (Figure 3A). The carbohydrate chains from the intracellular Gp3 were completely cleaved by both these enzymes, confirming that Gp3-HA is retained in ER or a *cis*-Golgi compartment, as Endo H cannot cleave the *N*-glycans with additional modifications occurring in the medial or *trans*-Golgi. Therefore, the Gp3-HA expressed in infected cells exhibits the same behavior as in previously published data for the wild-type Gp3 [13,20,22].

### 3.5. The EAVGp3-HA Virions Include Gp3-HA in the Monomer and Gp2/Gp3-HA/Gp4 Trimer Form

To demonstrate that the tagged Gp3-HA is incorporated into the virions, the BHK-21 cells were infected with the P1 EAVGp3-HA stock. After 22 h post-infection, the supernatants were collected, centrifuged at low speed, and purified using filter devices. A major part of the purified virus sample (95%) was subjected to immunoprecipitation (IP) with anti-HA antibodies, and analyzed using SDS-PAGE under non-reducing and reducing conditions. As expected, in non-reducing conditions, the Gp3-HA was present in virions in small amounts and was barely visible without precipitation (Figure 3B). In the sample subjected to IP, Gp3-HA was present at a band of 35 kDa and a band of 70 kDa. These apparent molecular masses correspond to the Gp3-HA monomer and Gp2/Gp3-HA/Gp4 trimer. In reducing conditions, only the monomeric form was present, suggesting that the multimeric form detected in the virions in the non-reducing condition is because of the covalent linkage of Gp3-HA to other minor proteins of EAV. Moreover, additional bands were present in the mock samples, which is probably because of IP and western blotting being performed using the same antibody. Unfortunately, detection of Gp3-HA with different anti-HA antibody in western blotting failed (mouse monoclonal anti-HA antibody, Enzo). In reducing conditions, the 50 kDa band is possible an antibody heavy chain; however, in non-reducing conditions, the bands indicated with triangles are probably detected covalently bond light and heavy chains of the antibodies used in IP. It is possible, that those antibody-derived bands might hide some bands coming from the targeting protein or protein complex. Therefore, we can assume that at least monomeric and trimeric Gp3-HA was present in the virion, but other forms cannot be excluded.

To investigate if the E protein associated in a virion with the Gp2/Gp3-HA/Gp4, the immunoprecipitates obtained using anti-HA antibodies and purified supernatant lysate (WSL), which is the virion fraction, and the whole cell lysates (WCL) were subjected to reducing SDS-PAGE and western blotting using anti-E antibodies (Figure 3C). The E protein was expressed in infected cells (WCL) as a double band; however, in the purified virion, only one band was detected, which is a feature that has been previously observed [12]. However, in the sample immunoprecipitated with anti-HA antibodies, no E protein could be detected, indicating that the E protein does not form any covalent linkages with the Gp2/Gp3-HA/Gp4 trimer in virions.

### 3.6. Gp3-HA and E Protein Partially Co-Localize with Each Other but not with N Protein

We studied the co-localization of the Gp3-HA, E protein, and N protein in EAV-infected BHK-21 cells using immunofluorescence microscopy (18 h p.i.). However, the Gp3-HA partially co-localized with E and N proteins, and co-localization was higher in the case of E protein (Mander’s coefficient = 0.61 ± 0.07) compared to that of N protein (Mander’s coefficient = 0.37 ± 0.16) (Figure 4A,B). Furthermore, very little co-localization was observed between E and N proteins (Mander’s coefficient = 0.29 ± 0.07) (Figure 4C).

### 3.7. The Gp3-HA and E Protein Primarily Localize within ER and cis-Golgi but to a Lesser Extent with ERGIC

In the next step, we investigated the co-localization of Gp3-HA with the cellular compartment specific markers. The BHK-21 cells were infected with recombinant EAVGp3-HA virus, and the cells (18h p.i.) were fixed and subjected to double immunostaining with rabbit anti-HA antibodies and antibody against a particular cellular compartment. The localization of Gp3-HA was analyzed using antibodies directed against the following markers of the secretory pathway: protein disulfide-isomerase (PDI) for ER; ERGIC-53 for ER-Golgi intermediate compartment, ERGIC; and membrin for *cis*-Golgi. The Gp3-HA was co-localized with ER (Mander’s coefficient = 0.66 ± 0.1) and *cis*-Golgi (Mander’s coefficient = 0.61 ± 0.11) markers (Figure 5A,C), and, to a lesser extent, with the ERGIC compartment marker (Mander’s coefficient = 0.39 ± 0.03) (Figure 5B).

Similarly, we investigated the co-localization of the E protein with the same cellular compartment’s markers. BHK-21 cells, infected with EAVGp3-HA virus, were subjected to double immunostaining with rabbit anti-E antibodies and antibody against a particular cellular compartment. The E protein co-localized with PDI, marker of ER (Mander’s coefficient = 0.53 ± 0.12) (Figure 6A). There was some co-localization with the cis-Golgi marker membrin (Mander’s coefficient = 0.41 ± 0.08) (Figure 6C) and minimal co-localization within the ERGIC (Mander’s coefficient = 0.26 ± 0.08) (Figure 6B).

## 4. Discussion

In this study, we describe the successful design and characterization of recombinant EAV that have separated ORFs encoding minor structural glycoproteins Gp3 and Gp4, which were then used to derive recombinant EAV with HA-tagged Gp3. Tagging the proteins in viral context might be a good tool to study localization and protein–protein interactions, particularly if the antibodies against those proteins are difficult to obtain. In arteriviruses, tagging the structural proteins is difficult because of the overlapping ORFs and nested genome.

Previously, the separation of the overlapping ORFs in EAV genome was achieved only for ORF4 and 5, ORF 5 and 6, and for ORFs 4, 5, 6 [23]. Moreover, a small 9 aa epitope was successfully added to the N terminus of the M protein (ORF6); however, the overlapping sequences between major structural protein coding gens ORFs 4, 5, and 6 are smaller compared to the overlap between genes coding for minor virion proteins. In the abovementioned studies, the introduction into the EAV genome composed of additional 17–41 nucleotides and was stable up to P2. In our study, we successfully introduced 136 nucleotides between structural virion genes and tested the stability of the expression of the HA-tag, which lasted till at least P15. This is a very stable tag expression comparing bigger tags, such as mCherry and GFP, which were introduced in parts of the genome encoding non-structural proteins that had fluorescent protein expression declining after a few passages [24,25].

These infectious clones, pEAV211s3/4 and pEAV211Gp3-HA, will enable easier mutagenesis on the C-terminus of the Gp3 and N-terminus of the Gp4 as changing the nt sequence of one ORF in particular regions will not influence the coding sequence in the other ORF.

In this study, we demonstrated that the addition of HA-tag to the Gp3 did not affect virus infectivity and replication. Gp3-HA was incorporated into the virions and was forming a multimeric structure, corresponding to the size of the Gp2/3/4 trimer. The trimer was not detected in infected cells, only in the virion, thus supporting earlier experiments [11,13]. The recombinant EAVGp3-HA will enable analysis of the entry of EAV as the information on the mechanisms of fusion and uncoating of the arteriviruses is still missing. Furthermore, the tagged protein could be used in proteomic experiments that explore host–virus interactions [26]. Information that the EAV genome supports a stable tag in the structural protein could be useful for vaccine development of arterivirus, e.g., in the techniques of virion purification or marker vaccine development [27].

In this study, we used the recombinant EAV to study the localization of the Gp3-HA. Previously, the localization of the Gp3 was tested in transfected cells only, and exclusively with the ER marker [13]. As the incorporation of minor proteins is E dependent we also performed detailed co-localization of the E. Clearly, these membrane proteins poorly co-localized with N, which was shown to be primarily located in the cytoplasm (inside DMVs) [28]. The Gp3-HA and E co-localized to some extent with each other, however we expected higher co-localization of those minor EAV proteins. Both Gp3-HA and E localized primarily in the ER, and some of the Gp3-HA was also found in the *cis*-Golgi compartment. The E protein co-localized with *cis*-Golgi to a lesser extent, then Gp3-HA, which can explain why the co-localization of the Gp3-HA and E was lower than expected.

Our study suggests that only a small fraction of the expressed Gp3-HA and an even smaller fraction of E protein was transported to the Golgi apparatus. The Gp3-HA and E did not co-localize in ERGIC; however, this does not rule it out as a budding site of Arteriviruses. Although the Gp3 does not contain any known ER retention motif, a majority of the protein was retained in the ER in our study. The *N*-glycans of Gp3-HA were sensitive to endoglycosidase H, confirming that most of the protein remains in the ER. We observed some localization of Gp3-HA and E in the cis-Golgi, possibly because some of the Gp3-HA and E might have escaped from ER because of overexpression and was transported back from the Golgi to ER [29].

Although expressed in high amounts in infected cells, only a small fraction of the E and minor glycoproteins Gp2, Gp3, and Gp4 end up in the virions [6,13]. A few reasons for this phenomenon are as follows. First, minor proteins might not be present at the assembly site. If Gp5/M, which is indispensable for VLPs formation, is located primarily in the Golgi, it could be the main budding site. In our study, we have shown that only some fraction of Gp3-HA is also located in the *cis*-Golgi, while a majority of it seems to be retained in ER. Only properly folded and assembled transmembrane proteins are exported from the endoplasmic reticulum (ER) [29]. It is possible that the complex Gp2/Gp4 dimer, as well as Gp3 and E, moved to the Golgi, while monomers remained in ER. Unfortunately, the Gp3-HA does not form a disulfide-linked trimer in the cell; therefore, we could not perform the fractionation of internal membranes to see if the Gp3-HA did move from the ER to the Golgi on oligomerization with Gp2 and Gp4. Disulfide-linked trimer formation is mediated after virion release, preferably at more basic pH [11]. We also detected the higher forms of Gp3-HA in non-reducing conditions only in virions.

The site of arterivirus assembly is still not fully understood. Previous electron microscopy studies suggest budding from intracellular membranes: ER or Golgi [14,15]. Clearly, for the production of VLPs, the nucleocapsid N coupled with RNA and the Gp5/M heterodimer formation is essential [6,30]. This is in contrast to the assembly of *Coronaviridae* (which are in the same order as *Arteriviridae*—Nidovirales), in which only the M and E proteins drive virion budding [31]. The failure to achieve budding by co-expression of only N, Gp5, and M without the EAV RNA suggests that there are additional factors in the EAV particle formation. It is possible that the arterivirus assembly is coupled to viral genome replication, interaction of RNA with structural proteins, or to the expression of non-structural proteins.

It has been shown primarily for other Nidovirales, such as the Coronaviruses, that the efficient incorporation of viral proteins into virions depends on two determinants, i.e., protein trafficking and interaction between proteins at the budding site, which in the case of Coronaviruses is ERGIC [32]. However, the factors that control the site of budding in well-studied Coronaviruses are unknown. When expressed independently or during an infection, many coronaviral proteins pass through the ERGIC and localize in the Golgi complex, e.g., the spike S protein is expressed even on the plasma membrane [33,34].

In Arteriviruses, Gp5/M dimers, which are essential for EAV budding, were shown to be located primarily in the Golgi apparatus (co-localization with mannosidase II), although at subsequent time points of the infection, the M was also present in the ER [35] and Golgi localization was not achieved if the dimer formation was blocked. If the budding occurs where the Gp5/M is localized, this could be the mechanism of lower incorporation into the virions of minor proteins, i.e., they are not abundant at the assembly site. In this study, we have shown that the Gp3-HA and E localized mainly in the ER. However, the mechanism of lower incorporation of the E and minor glycoproteins in Arteriviruses might be different, and not dependent on the localization of the proteins. In coronaviruses the E protein, which is essential for virion formation, remains at the budding site, but is incorporated into virions in small amounts, by unknown mechanism [36]. In the influenza A virus the M2 protein is excluded from the budozone, presumably because it localizes at the edges of the lipid rafts at plasma membrane, the site of Infuenza A budding [37,38]. Further experiments are needed to explain lower incorporation of the minor proteins into EAV virions.

## 5. Conclusions

In this study, we successfully introduced the HA-tag into the minor structural protein Gp3 of the EAV. The tagged protein was incorporated into the virion particles, whereas the mutant virus behaved similar to the WT. The tagged Gp3 in the virus context might facilitate research on the biology of EAV, e.g., in this study, we analyzed the localization of the Gp3-HA in the infected cells.

The Gp3-HA primarily localized in the ER, some Gp3-HA was present in cis-Golgi, but very little was present in the ERGIC.

This study shows that the Arterivirus genome tolerates substantial manipulation within genes coding for structural proteins that are responsible for cellular tropism without EAV virus infectivity loss. The separation of ORF3 and ORF4 in reverse genetic plasmids can facilitate mutagenesis on the terminal parts of Gp3 and Gp4. These results can be also implemented for the development of new arteriviral vaccines.

## Figures and Tables

**Figure 1 viruses-11-00735-f001:**
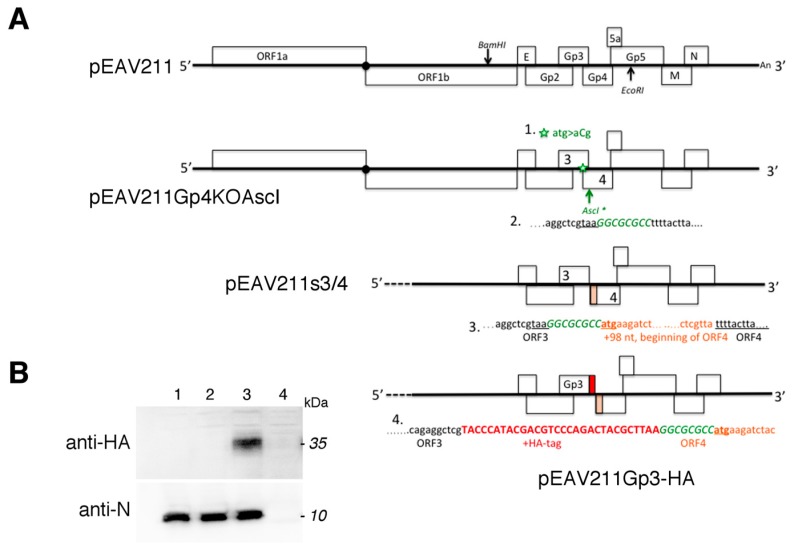
Schematic of the recombinant equine arteritis virus (EAV) construction (**A**). Introduction of the ORF4 Gp4 start codon mutation (1, green star) and the *AscI* nucleotide sequence (2, arrow). Separation of the overlapping ORF3 and ORF4 by introducing the beginning of the Gp4 sequence (3). Addition of the HA-tag sequence to the end of ORF3 (4). Expression of Gp3-HA and N protein in BHK-21 cells (**B**). Cells were transfected with in vitro-transcribed RNA produced from pEAV211, pEAV211s3/4, and pEAV211Gp3-HA. After the appearance of the cytopathic effect, the cells were lysed and subjected to SDS-PAGE and western blotting with anti-HA and anti-N antibodies. The cells were then transfected with in vitro-transcribed RNA: pEAV211 (WT), lane 1; pEAV211s3/4, lane 2; pEAV211Gp3-HA, lane 3; and untransfected cells, lane 4. The apparent molecular masses in kDa are shown on the right side. Expression of Gp3-HA and N in infected BHK-21 cells (**C**). Cells were infected with culture supernatants from BHK-1 cells previously transfected with pEAV211 (WT), pEAV211s3/4, and pEAV211Gp3-HA. They were then subjected to immunofluorescence 24 h post-infection with anti-N and anti-HA antibodies. WT—cells infected with EAV (wt); EAV-s3/4—cells infected with recombinant EAVs3/4 virus with separated ORFs 3 and 4; EAVGp3-HA—cells infected with recombinant EAVGp3-HA virus with HA-tagged Gp3; mock—uninfected cells. N is shown in green, Gp3-HA is shown in red, and DAPI—cell nuclei. Scale bar = 10 µm.

**Figure 2 viruses-11-00735-f002:**
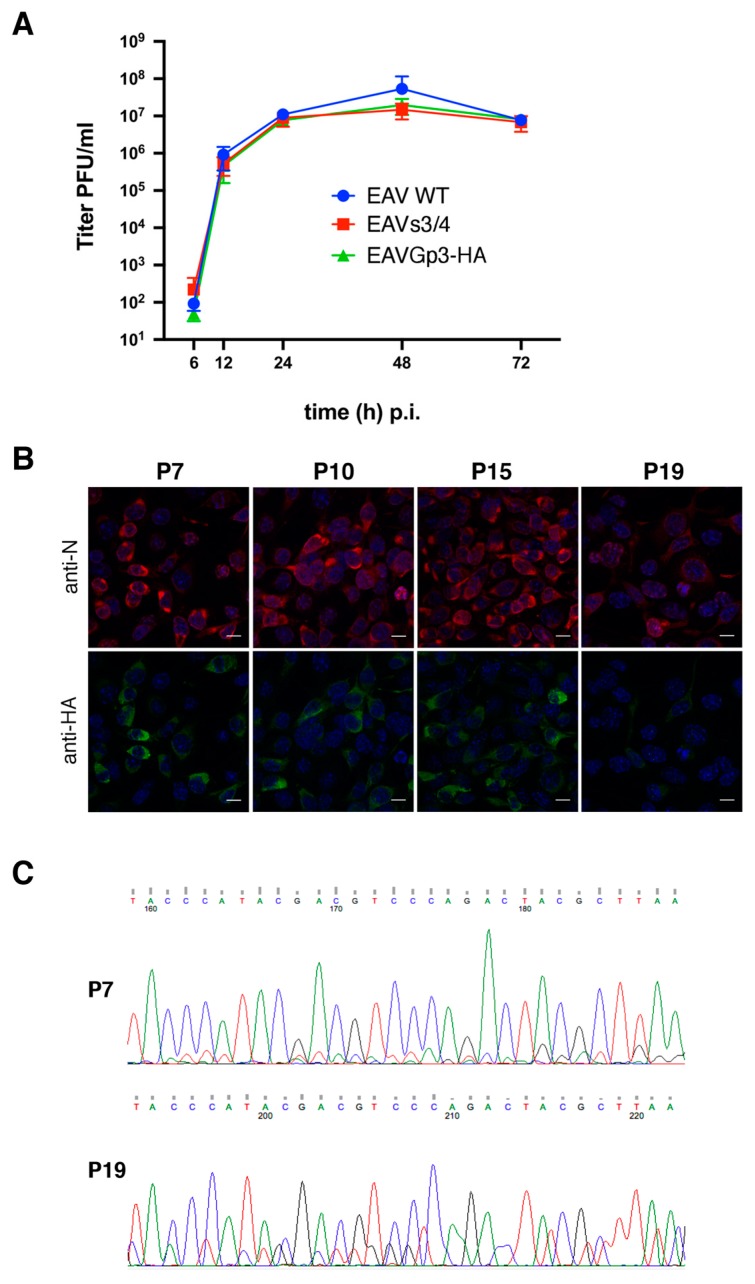
Growth kinetics of the EAV recombinant viruses (**A**). BHK-21 cells were infected with an MOI of 0.1 of P1 stocks of EAV (wt) and recombinant EAVs3/4 and EAVGp3-HA viruses. Supernatants were collected at 6, 12, 24, 48 and 72 h post-infection and subjected to plaque assay. PFU/mL—plaque forming unit per mL. Expression of Gp3-HA is stable upon infection (**B**). EAVGp3-HA virus was passaged 19 times on fresh BHK-21 cells. Cells were infected with culture supernatants containing recombinant viruses from passages P7, P10, P15, and P19, and subjected to immunofluorescence assay with anti-HA-tag and anti-N antibodies after 18 h p.i. N is shown in red and Gp3-HA is shown in green. Cell nuclei were stained with DAPI. Chromatograms of viral DNA sequences obtained from EAVGp3-HA infected cells (**C**). Cells infected with P7 and P19 were subjected to RT-PCR. The fragment of the HA-tag nt sequence is shown. The consensus sequence was maintained, but the presence of additional nt peaks in P19 suggested the increase of quasispecies of the virus.

**Figure 3 viruses-11-00735-f003:**
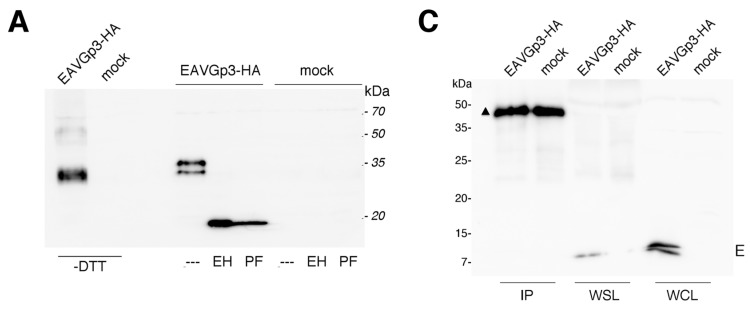
Characterization of Gp3-HA in infected cells (**A**). BHK-21 cells were infected with P1 EAVGp3-HA at an MOI of 1. Then, 18 h p.i. cells were washed, scraped from the dish, pelleted, and lysed in a non-reducing buffer (-DTT; left part of the blot) or in the deglycosylation buffer (right part of the blot). To analyze if the Gp3-HA is *N*-glycosylated in infected cells, a part of the samples was subjected to deglycosylation with endoglucosidase H (EH) or PNgase F (PF). The samples were then subjected to SDS-PAGE and western blotting with anti-HA antibodies. EAVGp3-HA - cells infected with EAVGp3-HA; mock - uninfected cells. The apparent molecular masses in kDa are shown on the right. Gp3-HA is present in the virions as a monomer and a trimer (**B**). To investigate whether the tagged Gp3-HA is incorporated into the virions, the BHK-21 cells were infected with P1 EAVGp3-HA stock at an MOI of 1. The supernatants (22 h p.i.) were collected, cleared with low-speed centrifugation, and concentrated on the filter devices. Major part (95%) of the purified virus sample was subjected to immunoprecipitation (IP) with anti-HA antibodies, and the remaining 5% of the sample was boiled with non-reducing or reducing SDS buffer (WSL—whole supernatant lysate). All samples were subjected to SDS-PAGE under non-reducing and reducing conditions and to western blotting with anti-HA antibodies. EAVGp3-HA—cells infected with EAVGp3-HA; Mock—uninfected cells. An asterisk indicates the monomeric Gp3-HA, whereas the three stars indicate the trimer Gp2/Gp3-HA/Gp4. Triangle indicates probable antibody chains. HC—heavy chain of antibody used in IP. The apparent molecular masses are shown in kDa. Gp3-HA does not form covalently bond complex with E in a virion (**C**). To investigate whether the Gp3-HA forms a complex with E, the samples described in B were subjected to SDS-PAGE in reducing conditions and western blotting with anti-E antibodies. IP—samples that were immunoprecipitated with anti-HA antibodies. WSL—whole supernatant lysate. Moreover, the infected cells were washed, pelleted, and lysed in a reducing buffer (WCL—whole cell lysate). Mock—uninfected cells. The molecular weight is shown in kDa.

**Figure 4 viruses-11-00735-f004:**
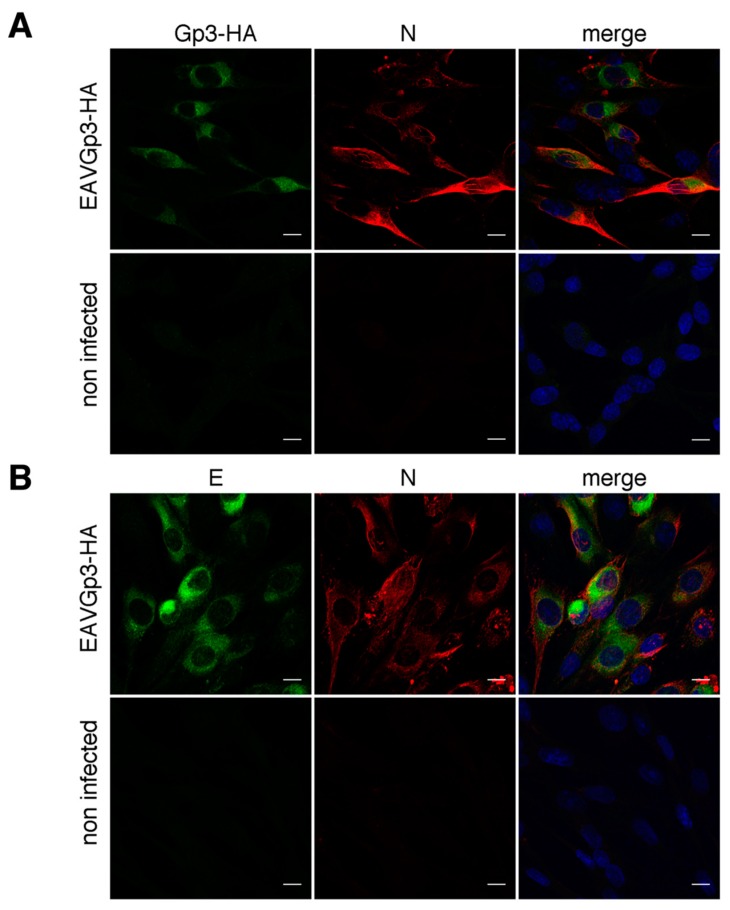
Gp3-HA partially co-localize with E protein (**A**). The BHK-21 cells were infected with EAVGp3-HA at an MOI of 1 or left uninfected. Then, the cells (18 h p.i.) were subjected to immunofluorescence with mouse anti-HA antibodies and rabbit anti-E antibodies and secondary Alexa Fluor 568-conjugated anti-mouse antibodies and Alexa Fluor 488-conjugated anti-rabbit antibodies, respectively. E is shown in green, Gp3-HA is shown in red. Gp3-HA shows little co-localization with N protein (**B**). Cells were infected as above; Cells (18 h p.i.) were then subjected to immunofluorescence with rabbit anti-HA antibodies and mouse anti-N antibodies and secondary Alexa Fluor 568-conjugated anti-mouse antibodies and Alexa Fluor 488-conjugated anti-rabbit antibodies, respectively. N is shown in red, Gp3-HA is shown in green. E shows little co-localization with N protein (**C**). Cells were infected as above; Cells (18 h p.i.) were then subjected to immunofluorescence with rabbit anti-E antibodies and mouse anti-N antibodies and secondary Alexa Fluor 568-conjugated anti-mouse antibodies and Alexa Fluor 488-conjugated anti-rabbit antibodies, respectively. N is shown in red, E is shown in green. The cell nuclei were stained with DAPI. Scale bar = 10 µm.

**Figure 5 viruses-11-00735-f005:**
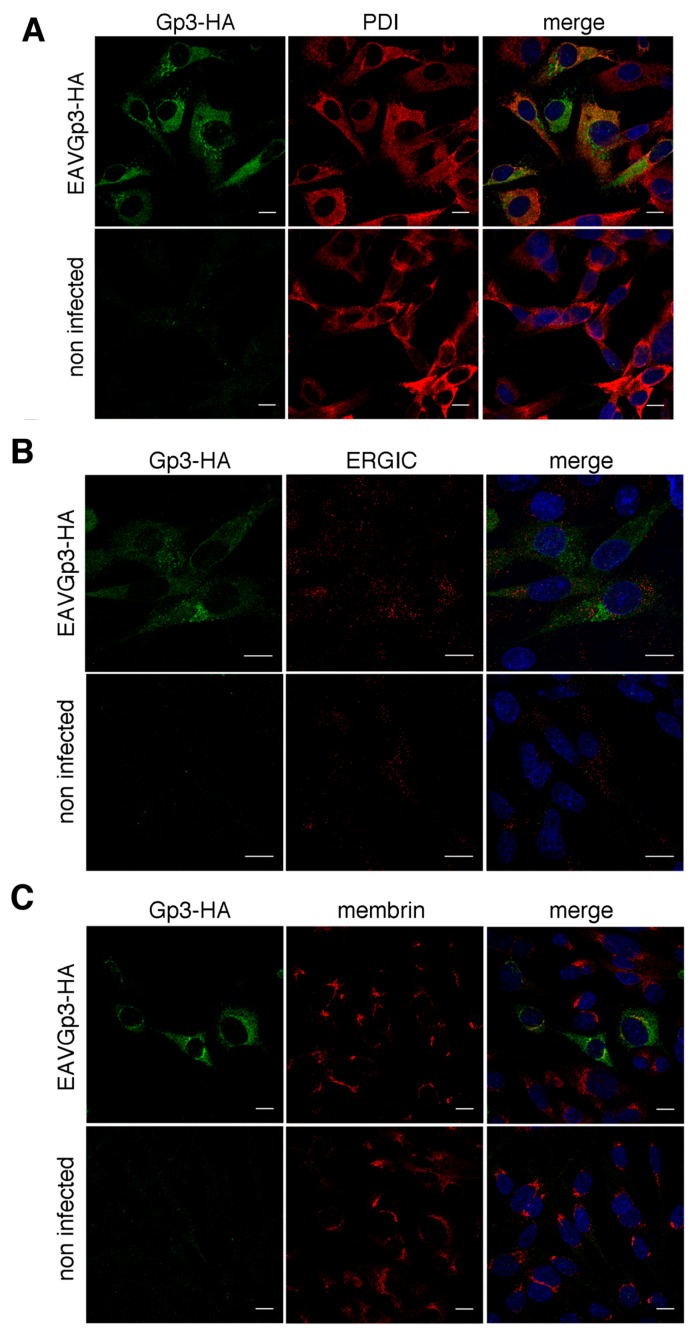
Co-localization of Gp3-HA with markers of the secretory pathway. The BHK-21 cells were infected with EAVGp3-HA at an MOI of 1 or left uninfected. Cells (18 h p.i.) were then subjected to immunofluorescence. Gp3-HA co-localized with PDI, marker for ER (**A**). Confocal analysis of the immunofluorescence samples stained with rabbit anti-HA antibodies and mouse anti-PDI antibodies and secondary Alexa Fluor 568-conjugated anti-mouse antibody and Alexa Fluor 488-conjugated anti-rabbit antibodies, respectively. Gp3-HA is shown in green, PDI is shown in red. Gp3-HA minimally co-localized with ERGIC (**B**). Confocal analysis of the immunofluorescence samples stained with rabbit anti-HA antibodies and mouse anti-ERGIC53 antibodies and secondary antibodies Alexa Fluor 568 anti-mouse and Alexa Fluor 488 anti-rabbit antibodies, respectively. Gp3-HA is shown in green, ERGIC is shown in red. Gp3-HA co-localizes with membrin, marker for cis-Golgi (**C**). Confocal analysis of the immunofluorescence samples stained with rabbit anti-HA antibodies and mouse anti-membrin antibodies and secondary Alexa Fluor 568-conjugated anti-mouse antibody and Alexa Fluor 488-conjugated anti-rabbit antibodies, respectively. Gp3-HA is shown in green, membrin is shown in red. The cell nuclei were stained with DAPI. Scale bar = 10 µm.

**Figure 6 viruses-11-00735-f006:**
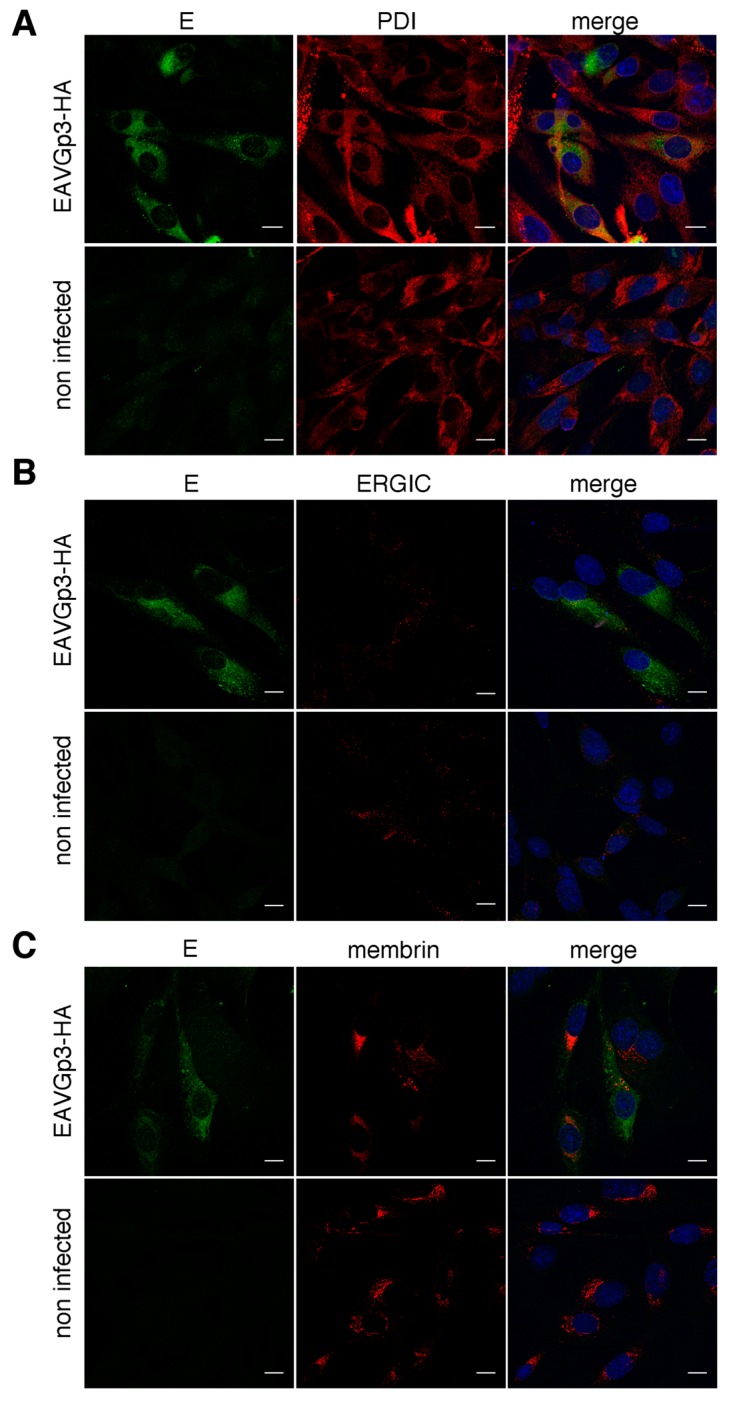
Co-localization of E with markers of the secretory pathway. The BHK-21 cells were infected with EAVGp3-HA at an MOI of 1 or left uninfected. The cells (18 h p.i.) were then subjected to immunofluorescence. E partially co-localizes with PDI, marker for ER (**A**). Confocal analysis of the immunofluorescence samples stained with rabbit anti-E antibodies and mouse anti-PDI antibodies and secondary antibodies Alexa Fluor 568 anti-mouse and Alexa Fluor 488 anti-rabbit antibodies. E is shown in green, PDI is shown in red. E minimally co-localizes with ERGIC (**B**). Confocal analysis of the immunofluorescence samples stained with rabbit anti-E antibodies and mouse anti-ERGIC53 antibody and secondary antibodies Alexa Fluor 568 anti-mouse and Alexa Fluor 488 anti-rabbit antibodies, respectively. E is shown in green, ERGIC is shown in red. E partially co-localizes with membrin, marker for cis-Golgi (**C**). Confocal analysis of the immunofluorescence samples stained with rabbit anti-E antibodies and mouse anti-membrin antibodies and secondary antibodies Alexa Fluor 568 anti-mouse and Alexa Fluor 488 anti-rabbit antibodies. E is shown in green, membrin is shown in red. The cell nuclei were stained with DAPI. Scale bar = 10 µm.

**Table 1 viruses-11-00735-t001:** Oligonucleotides used for this study.

Primer Name	Sequence 5′ > 3′	Plasmid Generated or Purpose
RGEAVBamHIIFor	gcttacggatcccacttcatcttttccc	Primer for cloning into pEAV211
RGEAVEcoRIRev	ggtggtgaattcacggccatagtaaataaaag	Primer for cloning into pEAV211
Gp4KOFor	ctttttcctttgtagacgaagatctacggctg	pEAV211Gp4KO
Gp4KORev	cagccgtagatcttcgtctacaaaggaaaaag	pEAV211Gp4KO
EAV211AsciFor	ctcgtaaggcgcgccttttacttacattagtc	pEAV2114KOAscI
EAV211AsciRev	gtaaaaggcgcgccttacgagcctctgcag	pEAV2114KOAscI
ReconAsciGp4For	gcttacggcgcgccatgaagatctacggctgc	pEAV211s3/4
EAVGp3-HARev	Gatcggcgcgccttaagcgtagtctgggacgtcgtatgggtacgagcctctgcagcgtg	pEAV211Gp3-HA
EAVFor	gtgaatgtctttgctaatg	RT-PCR primer
EAVRev	ccagaagtaaacaatgag	RT-PCR primer

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
