# Peer review of "Production of Recombinant EAV with Tagged Structural Protein Gp3 to Study Artervirus Minor Protein Localization in Infected Cells"

_viruses, 2019, doi:10.3390/v11080735_

Round 1

Reviewer 1 Report

The manuscript “viruses-555803” entitled “Production of recombinant EAV with tagged structural protein Gp3 to study Artervirus minor protein localization in infected cells.” is a well-designed and well written. The authors have successfully introduced the HA-tag into the minor structural protein Gp3 of the Equine arteritis virus (EAV). The tagged protein was incorporated into the virion particles, whereas the mutant virus behaved similar to the WT. The tagged Gp3 in the virus context might facilitate research on the biology of EAV. The Gp3-HA primarily localized in the ER, some Gp3-HA was present in cis-Golgi, but very little was present in the ERGIC. This study shows that the Arterivirus genome tolerates substantial manipulation within genes coding for structural proteins that are responsible for cellular tropism without EAV virus infectivity loss. The separation of ORF3 and ORF4 in reverse genetic plasmids can facilitate mutagenesis on the terminal parts of Gp3 and Gp4. Interestingly, results of this study can be also implemented for the development of new arteriviral vaccines. Generally, the English language of the manuscript is adequate; the quality of the figure and table is satisfactory, the reference list cover the relevant literature adequately and in an objective manner. However these minor points before publication:

1. Line 15/Page 1: the description of the abbreviation EAV should be Introduced “Equine arteritis virus (EAV)”.

2. Line 105/Page 3: Table (1S) should be checked for the language “e.g.  purpouse” and be moved from the supplementary into the main article “Table (1)”.

Author Response

We thank the reviewer's for they work.

 Line 15/Page 1: the description of the abbreviation EAV should be Introduced “Equine arteritis virus (EAV)”.

The description was introduced to the beginning of the abstract.

2. Line 105/Page 3: Table (1S) should be checked for the language “e.g.  purpouse” and be moved from the supplementary into the main article “Table (1)”.

Table S1 was moved to the main article (line 112), the spellcheck was performed.

Reviewer 2 Report

In the study entitled “Production of recombinant EAV with tagged structural protein Gp3 to study Artervirus minor protein localization in infected cells” , Anna et a.,  generated two EAV recombinants : one recombinant separately expressing ORF3 and ORF4 and the other incorporated with a HA tag after ORF3. Both recombinants were successfully rescued and were stable on passaging and replicated in titers similar to the wild-type EAV. The expression and localization of the Gp3-HA protein was further identified with the recovered recombinant virus. The recombinant generated in this study can be an important research tool for the future marker vaccine development as well as pathogenesis mechanism study for EAV. The study is well performed and presented, but there are still some minor comments and technique questions need to be addressed: i. for viruses supernatant collected for western-blot, the authors used a filter for virus concentration, but not described in detail, it is necessary to give more information and details in the material and methods section. Meanwhile, why not using cell lysates instead of supernatant, since there are still serum in the supernatant (not sure whether the concentrated virus are free of serum, so that’s why more details should be given for the concentration of virus). On the other hand, the concentrated virus supernatant might even have lower virus copies compared to the cell lysates, it might be the reason the protein is almost undetachable in the whole supernatant lysate.  ii, for IP experiment, the author used the same antibody for both IP and western-blot, there are possibilities that those bands coming from the chains of antibodies might hide some bands coming from the targeting protein or protein complex. So it should be careful about the statement from this results. iii,I suppose it might be typo in page 19 Line 509 “lover then”. In overall, from my point of view, the study is well written and discussed, it is acceptable to be published in this journal after minor revision.

Author Response

We thank the reviewer for his or her work.

i. for viruses supernatant collected for western-blot, the authors used a filter for virus concentration, but not described in detail, it is necessary to give more information and details in the material and methods section. Meanwhile, why not using cell lysates instead of supernatant, since there are still serum in the supernatant (not sure whether the concentrated virus are free of serum, so that’s why more details should be given for the concentration of virus). On the other hand, the concentrated virus supernatant might even have lower virus copies compared to the cell lysates, it might be the reason the protein is almost undetachable in the whole supernatant lysate.

Additional information about virus concentration was added to material and methods section (Lines 233-239). The supernatant contained 1% of FCS, as addition of FCS generally gives higher virus titers. 

We used the supernatants and not cell lysate to make sure that we will detect Gp3-HA as a structural virion protein, and not just Gp3-HA expressed in infected cells. It was shown previously that the Gp3 forms a trimer with Gp2 and Gp4 only in a virion. Additionally, very little of the minor proteins are present in virion, despite substantial expression in the infected cells. We wanted to see if the addition of the HA tag to the Gp3 does not interfere with protein incorporation into EAV virions and with trimer formation.  The reason why the protein is almost undetectable in whole supernatant lysate is probably the proportion, in WSL its 2.5 % of all sample (while in IP it is the remaining supernatant). 

ii, for IP experiment, the author used the same antibody for both IP and western-blot, there are possibilities that those bands coming from the chains of antibodies might hide some bands coming from the targeting protein or protein complex. So it should be careful about the statement from this results.

For the WB after IP we tried the other anti-HA antibody (mouse monoclonal anti-HA tag antibody (Enzo Life Sciences, USA), but no bands were detected. This information was added to result section lines 398-399:             "Unfortunately, detection of Gp3-HA with different anti-HA antibody in western blotting failed (mouse monoclonal anti-HA antibody, Enzo). "

Moreover we added statement about IP results, lines 402-405       "It is possible, that those antibody-derived bands might hide some bands coming from the targeting protein or protein complex. Therefore, we can assume that at least monomeric and trimeric Gp3-HA was present in the virion, but other forms cannot be excluded. "

iii,I suppose it might be typo in page 19 Line 509 “lover then”. 

Word has been corrected to "lower"